# A conserved motif suggests a common origin for a group of proteins involved in the cell division of Gram-positive bacteria

**Mikel Martinez-Goikoetxea**[ORCID]**, Andrei N. Lupas***

Department of Protein Evolution, Max Planck Institute for Developmental Biology, Tübingen, Germany

* andrei.lupas@tuebingen.mpg.de

## Abstract

DivIVA, GpsB, FilP, and Scy are all involved in bacterial cell division. They have been reported to interact with each other, and although they have been the subject of considerable research interest, not much is known about the molecular basis for their biological activity. Although they show great variability in taxonomic occurrence, phenotypic profile, and molecular properties, we find that they nevertheless share a conserved N-terminal sequence motif, which points to a common evolutionary origin. The motif always occurs N-terminally to a coiled-coil helix that mediates dimerization. We define the motif and coiled coil jointly as a new domain, which we name DivIVA-like. In a large-scale survey of this domain in the protein sequence database, we identify a new family of proteins potentially involved in cell division, whose members, unlike all other DivIVA-like proteins, have between 2 and 8 copies of the domain in tandem. AlphaFold models indicate that the domains in these proteins assemble within a single chain, therefore not mediating dimerization.

**Data Availability Statement:** All relevant data are within the paper, its Supporting information file,

## Introduction

DivIVA proteins localize to negatively-curved membranes, such as the cell poles and the division septum, to which they recruit various proteins, depending on the organism and the physiological conditions [1]. Homologs of this protein are found predominantly in Gram-positive bacteria, but have also been reported in other major clades of bacteria. GpsB, a paralog of DivIVA in Firmicutes, is thought to complement the role of DivIVA in coordinating peptidoglycan synthesis at sites of cell division and elongation [2, 3]. In contrast to DivIVA and GpsB, FilP is considered to be primarily a cytoskeletal protein, but has also been implicated in cell division [4]. Thus, in Streptomyces species, FilP forms a concentration gradient that increases towards the cell poles, and is thought to contribute to the polar localization of DivIVA [5]. Scy (Streptomyces CYtoskeletal protein) is a paralog of FilP [6], found in Streptomyces species only. It is the largest protein in this set at about 4 times the size of FilP and has been reported to interact with both FilP and DivIVA [7]. All these proteins form dimeric coiled coils, differing in the length and periodicity of their coiled-coil segments.

and a Mendeley repository available at https://data.mendeley.com/datasets/bn627zbymx.

**Funding:** The authors were supported by institutional funds of the Max Planck Society. The funders had no role in study design, data collection and analysis, decision to publish, or preparation of the manuscript.

**Competing interests:** The authors have declared that no competing interests exist.

Coiled coils are a widespread supersecondary structure motif that consists of two or more α-helices that wind around a central axis and interlock their side-chains systematically along the core of the structure [8]. Their packing geometry, widely considered the hallmark of coiled coils, is known as knobs-into-holes and is achieved by packing a given core residue (knob) into a cavity formed by 4 residues on a symmetry-related α-helix (hole) [9]. Because of their different solvent accesibility, core residues tend to be hydrophobic, while all other residues of the coiled coil are typically polar. In nature, coiled coils are primarily built by sequence repeats of seven residues (heptads), featuring hydrophobic residues at the first and fourth positions. Using the general nomenclature of coiled coils, which denotes the seven positions of the repeat as *a-g*, the hydrophobic residues tend to be found in positions *a* and *d*. While heptads are the only repeat fully compatible with knobs-into-holes packing, hence, possibly, their dominance in nature, other sequence repeats may form coiled-coil structures. The most frequent of these are eleven-residue repeats (hendecads, labeled *a-k*, with hydrophobics predominantly in *a*, *d*, and *h*) and fifteen-residue repeats (pentadecads, labeled *a-o*, with hydrophobics predominantly in *a*, *d*, *h* and *l*). In all these proteins, the degree to which individual helices wind around the central axis of the bundle (supercoiling) is given by the difference between the sequence periodicity of hydrophobic residues and the structural periodicity of an undistorted α-helix. Thus, for example, the sequence periodicity of hydrophobic positions in heptads is 7/2 = 3.5, while the structural periodicity of an undistorted α-helix is 3.63 residues/turn. Therefore, the α-helix being a right-handed spiral, the difference of -0.13 specifies a left-handed supercoil. Correspondingly, the equivalent difference for hendecads is 0.03, specifying a very minor degree of supercoiling, and the difference for pentadecads is 0.12, specifying a right-handed supercoil of the same magnitude as the left-handed supercoil in heptads. Besides changing the degree of supercoiling, departures from heptad periodicity also affect the packing geometry. Particularly in hendecads, the different structural periodicity leads to one core residue per repeat pointing directly towards the central axis of the bundle, a packing mode referred to as knobs-to-knobs. The reduced distance between the side-chains results in a steric constraint, and, to avoid clashes, the distance must be increased by means of assembling into higher oligomeric states or with the core position featuring a small side-chain, particularly Alanine. Correspondingly, FilP and Scy, which are dimers but contain extended stretches of hendecad coiled coils, show a high proportion of Alanine residues in their hydrophobic cores [6].

Whereas heptad repeats have been studied extensively and can be predicted effectively from protein sequences, hendecads have hitherto remained largely unexplored. In order to produce a database of reliable hendecad sequences, we therefore searched for this periodicity by tandem repeat detection over the non-redundant protein sequence database (NR). In the process, we also encountered Scy and FilP, which, to our surprise, bore a remarkable similarity to DivIVA and GpsB in their N-terminal part. Here, we define the common motif in these four protein families and describe a fifth family that uniquely bears between 2 and 8 repetitions of this DivIVA-like motif. We propose that this family is yet another component of the Gram-positive cell-division machinery.

## Materials and methods

### 1 Discovery of the similarities between DivIVA, GpsB, FilP and Scy

As part of the sequence analysis of proteins detected in our survey for hendecads, we also analyzed representatives of Scy and FilP (WP_066885126.1 and WP_172158822.1) with HMMER (version 3.3, http://hmmer.org) [10] against the PFAM domain database (release 34.0, https://pfam.xfam.org) [11], and obtained intriguing matches between their N-terminal regions and the corresponding region in the DivIVA profile. Further HMM-HMM (Hidden Markov

Model) searches with HHpred against the PDB (Protein Data Bank) database in the MPI Bio-informatics Toolkit (https://toolkit.tuebingen.mpg.de/tools/hhpred, PDB_mmCIF70_14_Apr database using default parameters) [12] gave strong matches to DivIVA (PDB identifier 2WUJ), GpsB (4UG3), and Wag31 (6LFA). All matches were anchored by a set of highly conserved residues N-terminal to a dimeric heptad coiled coil.

## 2 Finding the conserved motif in DivIVA-like proteins

We collected homologs of DivIVA, GpsB, FilP and Scy using BLAST at NCBI (https://blast.ncbi.nlm.nih.gov) against the NR database (version May 2022) at the default E-value cutoff (0.05). In order to obtain a representative coverage of the protein families, we chose as starting sequences DivIVA from Bacillus (WP_163131312.1) and Streptomyces (WP_190537660.1), GpsB from Bacillus (NP_390100.1), and FilP and Scy from Streptomyces (GAU65504.1, MYV96918.1). We obtained 21171 sequences, which we then filtered with MMSeqs2 (https://toolkit.tuebingen.mpg.de/tools/mmseqs2) [13] to a maximum pairwise sequence identity of 80% and 80% minimum coverage, which yielded a total of 1816 sequences. In order to find conserved motifs in this dataset, we used MEME (version 5.4.1, https://meme-suite.org/meme/tools/meme) [14], whose highest scoring motif coincided with the conserved residues already identified in the HHpred searches. In order to ascertain the presence and conservation of this motif, we computed multiple sequence alignments (MSAs) for every representative sequence and their BLAST matches with ClustalOmega (version 1.2.4, http://www.clustal.org/omega) [15], and constructed logos using WebLogo (version 3.7.11, http://weblogo.threeplusone.com) [16]. Consensus sequences from the BLAST searches are shown in Fig 1C, and the logo produced by MEME for the highest scoring motif is shown in Fig 1D.

## 3 Construction of a cluster map of sequences with a DivIVA-like motif

In order to broadly collect sequences that feature this motif, we used the pattern GY[DN]xx[QE]V[DN] for searches of the NR30 database (NR database filtered to a maximum pairwise identity of 30%). In particular, we omitted the arginine in the MEME motif because we had noticed that some clades systematically contained a different residue at that position (E in actinobacterial DivIVA), and wanted to see whether other clades showed further diversity (indeed, the new family of DivIVA-like proteins we describe here shows a wide range of residues at this position, including a substantial proportion of Glycine residues). The pattern search yielded 5552 sequences. The relationships between these sequences were explored by clustering them according to their pairwise BLAST P-values in CLANS (https://toolkit.tuebingen.mpg.de/tools/clans) [17]. Clustering was done in default settings (attract = 10, repulse = 5, exponents = 1), and the map was imaged at the P-values given in the figure legends. In order to define clusters in the map, we constructed an undirected graph representation of the BLAST P-values at a threshold of 1E-15 and then used the Girvan-Newman algorithm as implemented in the NetworkX Python package (https://networkx.org) to automatically detect densely connected groups of sequences.

## 4 Detection and analysis of protein families containing DivIVA-like domains

In order to identify which of the sequences satisfying the pattern were part of protein families containing a DivIVA-like domain, we extended the pattern to include hydrophobic residues of the coiled-coil segment (GY[DN]xx[QE]V[DN]xx[ILV]xx[ILV]), and selected clusters in which at least half of the sequences had a match.

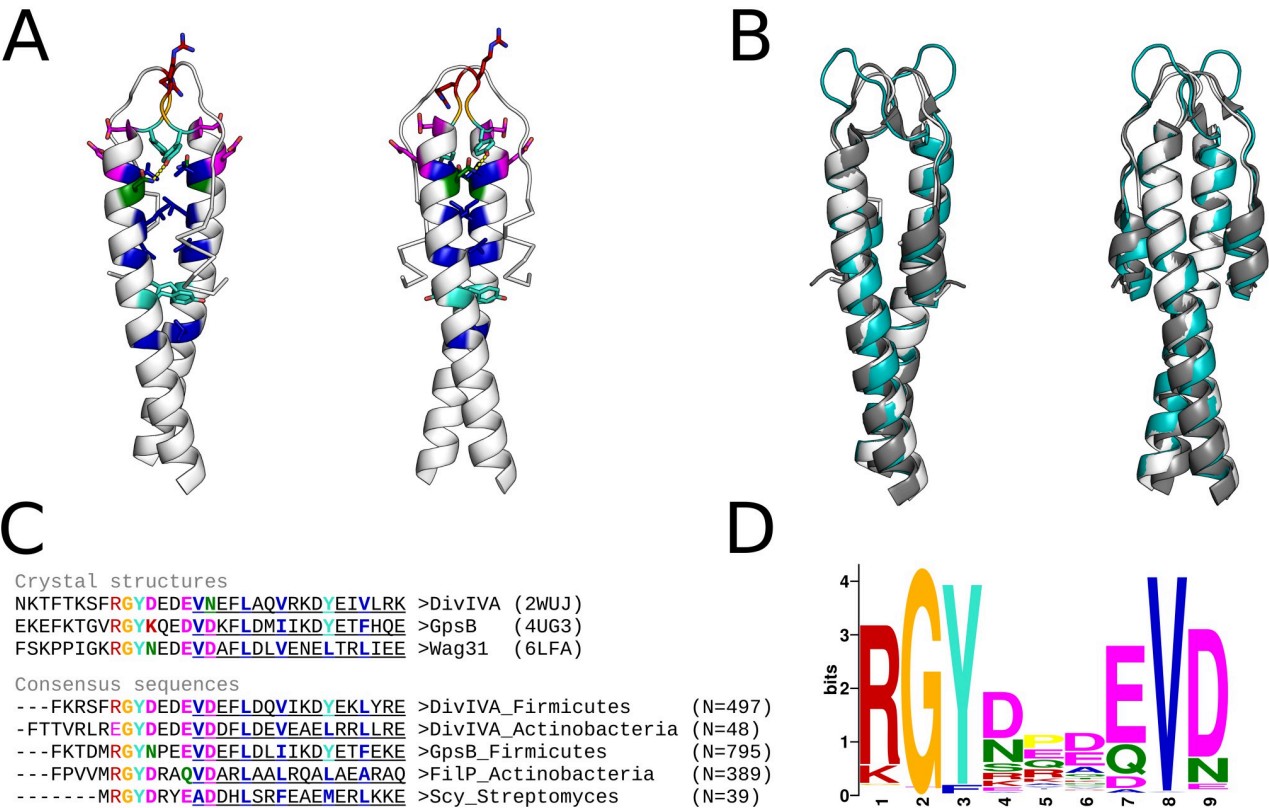

**Fig 1. Summary of the DivIVA-like domain.** (A) Cartoon renders of the DivIVA N-terminal domain (2WUJ); color-coded to match panel C, are the residues of the conserved motif as well as the core residues of the two following heptad coiled coils. (B) Superimposition of DivIVA (2WUJ, white), GpsB (4UG3, gray) and Wag31 (6LFA, teal); RMSD of the superimposition to 2WUJ is 1.0 and 2.2 Angstrom respectively. (C) Alignment of the sequences for the structures shown in panel B to the consensus sequences (N = number of sequences) of representatives for the major groups of DivIVA homologs (see Methods); the conserved motif and the core residues of the two following heptad coiled coils are highlighted in colors, and the Quick2D α-helical prediction is shown as underlined characters. (D) Logo representation of the top scoring motif found by MEME in the set of 1816 DivIVA homologs obtained as described in the Methods.

We analyzed these clusters for their taxonomic spectrum and sequence features, including interactive coiled-coil annotation with PCOILS and REPPER (https://toolkit.tuebingen.mpg.de/tools/repper) [18], disorder prediction with IUPRED2 [19], secondary structure prediction with Quick2D (https://toolkit.tuebingen.mpg.de/tools/quick2d) [12], and genomic context with GCsnap [20].

## 5 Structural predictions

We computed structural predictions for the PolyDIV proteins with AlphaFold (version 2.1.1) [21, 22]. First, we predicted monomers and homodimers for representative sequences of the PolyDIV clusters. Then, we produced artificial PolyDIV sequences by replacing the DivIVA-like domains of three PolyDIV proteins with DivIVA-like domains from either DivIVA, or GpsB, or FilP, for a total of 9 constructs (see S1 File). We predicted these both as monomers and as homodimers as well. Examples of the different topologies obtained are shown in Fig 3, and the complete set of sequences and models is provided in a Mendeley repository.

## Results

In a search for protein sequences with helical potential and hendecad periodicity, we encountered the Scy and FilP proteins, which have previously been described to contain hendecads [7]. In a scan for PFAM domains in these sequences (see Methods), we were surprised to obtain matches to the DivIVA domain in the N-terminal part of the proteins. Further, analysis of the N-terminal sequences by HMM-HMM comparison gave strong matches to the structures of DivIVA and its close homologs (Fig 1A and 1B), hinging on a pattern of conserved residues whose broad conservation we ascertained through multiple sequence alignments of the respective protein families. The consensus sequence for the conserved motif is RGYDxxEVD. In the crystal structures, residues RGYD are part of a loop that connects a short N-terminal helix to the coiled-coil helix mediating dimerization, which starts with EVD (Fig 1A–1C). In the motif, the G allows for a close contact between the symmetry-related loops of the dimer, the Y of one chain forms a hydrogen bond with the second D in the other chain, the first D is the N-terminal capping residue of the coiled coil, and the V is the first residue of its hydrophobic core. Based on these observations, we define the DivIVA-like domain as the conserved motif in the context of a coiled coil with heptad periodicity.

In order to collect as broadly as possible sequences featuring the DivIVA-like domain, we used pattern searches against sequence databases, as described in the Methods. We clustered the approximately 5500 sequences obtained in these searches by pairwise sequence similarity using CLANS. As is apparent from Fig 2A, the resulting cluster map consists of groups of well-connected sequences surrounded by a large halo of sequences that make few or no connections in the map. In order to separate the densely-connected groups into individual clusters, we used the Girvan-Newman algorithm for community detection. Finally, we checked the clusters for the presence of a hydrophobic pattern indicative of heptad coiled coils. This left us with the clusters highlighted in color in Fig 2A as the ones containing a DivIVA-like domain. At a P-value threshold of 1, these clusters came together at the center of the map (S1 Fig in S1 File). As we used more stringent thresholds, three main groups of clusters emerged. Mapping the known DivIVA-like proteins into the map, we identified one group as Scy/FilP (red group), and another one as DivIVA/GpsB (blue group). The third (purple) group did not have sequences that could be reliably mapped to any known DivIVA-like proteins.

In the DivIVA/GpsB group of clusters, the center is taken by DivIVA representatives from Firmicutes (lactobacillales, eubacteriales, clostridiales). Radiating outwards from the center of the map is GpsB (as expected, found only in Firmicutes), and two clusters of Firmicute proteins which are uncharacterized at present. Radiating inwards is the cluster of actinobacterial DivIVA proteins (from micrococcales, corynebacteriales and acidimicrobiales). All these sequences show essentially the same domain organization (Fig 2C), with variations in length and in the position and number of unstructured regions, as well as different preferences for the first residue of the motif ([R]GYD is the most common, followed by [E] and [G]). In the PDB structures of DivIVA-like proteins (Fig 1A), this residue protrudes from the top of the structure and may be part of an interacting surface, modulating binding partner specificity. In terms of genomic context, DivIVA proteins are found close to other proteins involved in cell division, such as SepF, FtsZ and FtsA, as well as YggT and YggS homologs. In turn, GpsB proteins are found close to RecU (Holliday junction resolvase), DnaD (component of the PriA primosome) and the penicillin-binding protein PBP1A.

The FilP/Scy group of clusters, as expected, contains exclusively actinobacterial sequences. The center is taken by FilP representatives (mainly from micrococcales, micromonosporales, and pseudonocardiales), as well as a few Scy sequences from Streptomyces (in Fig 2A and 2B, label 2). Radiating outwards are three clusters of uncharacterized proteins (from

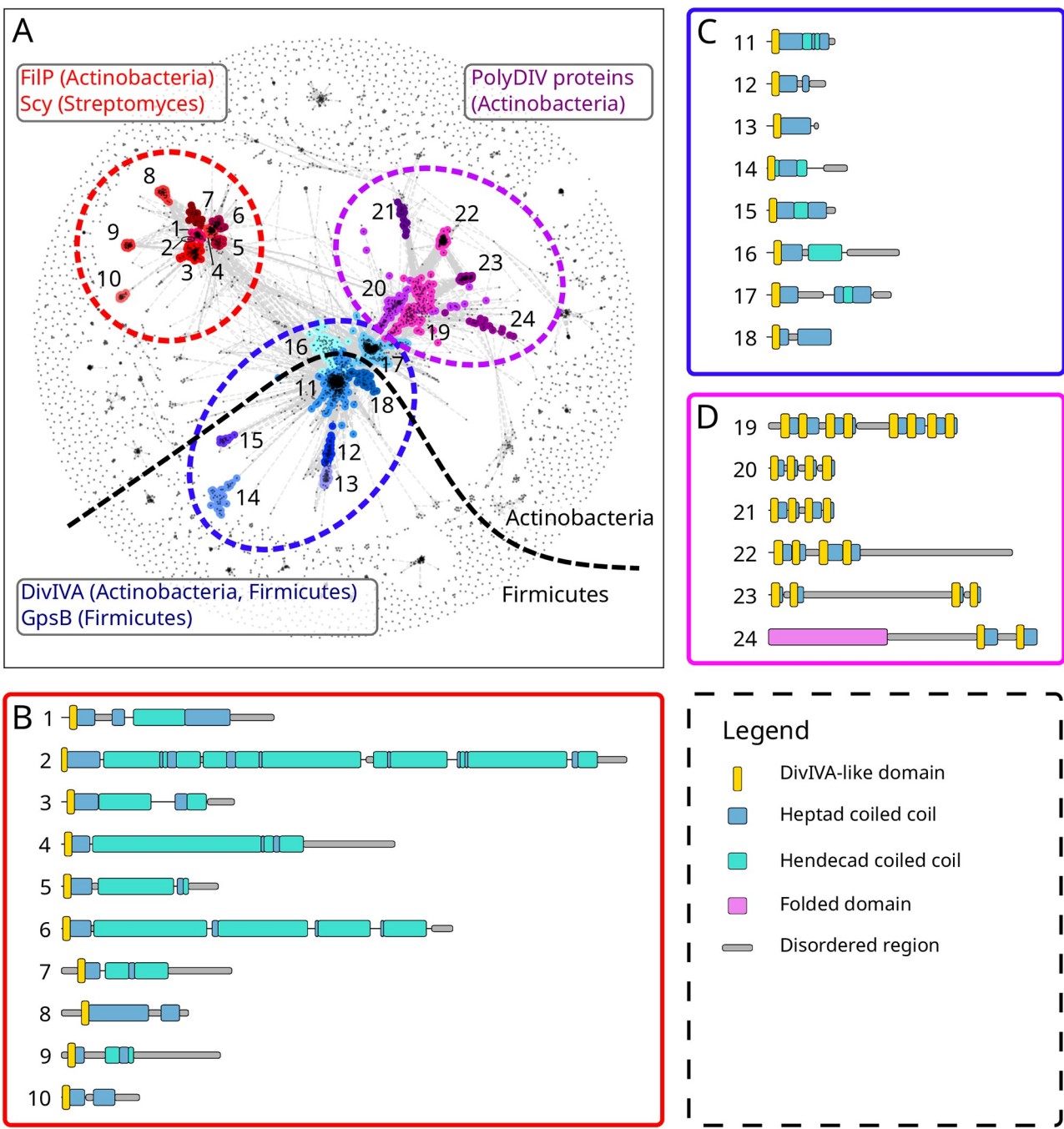

**Fig 2. A broad survey of DivIVA-like proteins.** (A) Cluster map of sequences with a match to the conserved motif in DivIVA-like proteins (see Methods). Clustering was done in 2D until equilibrium at a BLAST P-value of 1E-10. Connections represent similarities up to a P-value of 1E–10. The three groups of clusters that feature a DivIVA-like domain are colored and labeled. Cartoon representation of the sequence features for representative sequences of Scy/FilP (B), DivIVA/GpsB (C), and PolyDIV groups (D).

geodermatophilales, micromonosporales and pseudonocardiales). All the sequences in the FilP/Scy group show the same domain organization, but differ substantially in length, from under 200 residues to over 1000 (Fig 2B). All clusters show a preference for R before GYD, except for one of the uncharacterized clusters, where F or T are preferred. Surprisingly, Scy

sequences, which are closely embedded in the central FilP cluster, are N-terminally truncated and start right before the RGYD motif, thus lacking the N-terminal buttressing helix (Fig 1A). In terms of genomic context, there is a greater diversity than in the DivIVA/GpsB group. The Scy proteins as well as some FilP clusters are found close to crotonyl-CoA carboxylase/reductase and methylmalonyl-CoA epimerase genes, and in the case of Scy, also to a cellulose-binding protein, which by BLAST searches can be reliably identified as the FilP of Streptomyces species.

The third group of clusters also contains exclusively actinobacterial proteins (from micrococcales, streptosporangiales, pseudonocardiales, micromonosporales, corynebacteriales and bifidobacteriales). Its constituent sequences are distinctive in that they do not feature only one instance of the DivIVA-like domain, but rather several, ranging from a minimum of 2 to a maximum of 8, while the majority of the sequences feature 4 such domains. We have named this new, hitherto unknown group of proteins PolyDIV. Uniquely among DivIVA-like proteins, one cluster of PolyDIV proteins (no. 24 in Fig 2A and 2D) contains an additional folded domain, which is a Ser/Thr protein kinase. Most of the clusters in the PolyDIV group have a UMP kinase in their genomic vicinity.

The DivIVA-like domains in PolyDIV proteins tend to group in pairs, with connectors within the pairs being typically shorter than between pairs (Fig 2D). Although it is entirely possible that these proteins form dimers through the consecutive association of their DivIVA-like domains, their invariably even number and the length difference in their connectors suggested to us that the PolyDIV proteins might actually be monomeric, with consecutive DivIVA-like domains folding into DivIVA-like structures. Indeed, AlphaFold models predicted all representative PolyDIV proteins as monomers, with consecutive DivIVA-like domains assembling into DivIVA-like structures (first with second and third with fourth, as shown in Fig 3A). As a control, we also predicted the same proteins as homodimers; most models (7 of 9) retained the monomer topology, i.e. folding into DivIVA-like structures within a single chain (Fig 3C), while the rest showed a more complex topology with mixtures of intra- and inter-chain DivIVA-like structures (Fig 3D).

In order to probe the plausibility of intragenic amplification as a mechanism for the origin of PolyDIVs, we constructed synthetic PolyDIVs by amplification of DivIVA-like domains that are optimized for homodimeric interaction. To this end, we inserted the DivIVA-like domains of either DivIVA, or GpsB, or FilP into the linker frames of wild-type PolyDIV proteins (see Methods). When predicted as monomers, the majority of these synthetic PolyDIVs (8 of 9) displayed a topology identical to the wild-type PolyDIV monomers (Fig 3A), but when predicted as dimers, the majority (7 of 9) featured a more complex topology, combining intra- and inter-chain assembly (Fig 3D). These models show that intragenic amplification is sufficient to yield a monomeric PolyDIV topology from a precursor that is obligately homodimeric, but that further adaptations are needed to produce a structurally-specific variant.

## Discussion

We have shown that the DivIVA, GpsB, FilP and Scy proteins share a common N-terminal sequence motif (GYDxxEVD) followed by a short, dimeric heptad coiled coil. We define this combination as the DivIVA-like domain. A survey of proteins with DivIVA-like domains showed that these form three large groups of sequence clusters, the DivIVA/GpsB group, the FilP/Scy group, and a third, hitherto unreported group, which we name PolyDIV, for its most salient feature: unlike all other proteins containing the DivIVA-like domain, PolyDIVs contain the domain in multiple copies (2 to 8) within one chain. Among all these proteins, DivIVA is the one with the broadest taxonomic spectrum, as it is almost universal in Gram-positive

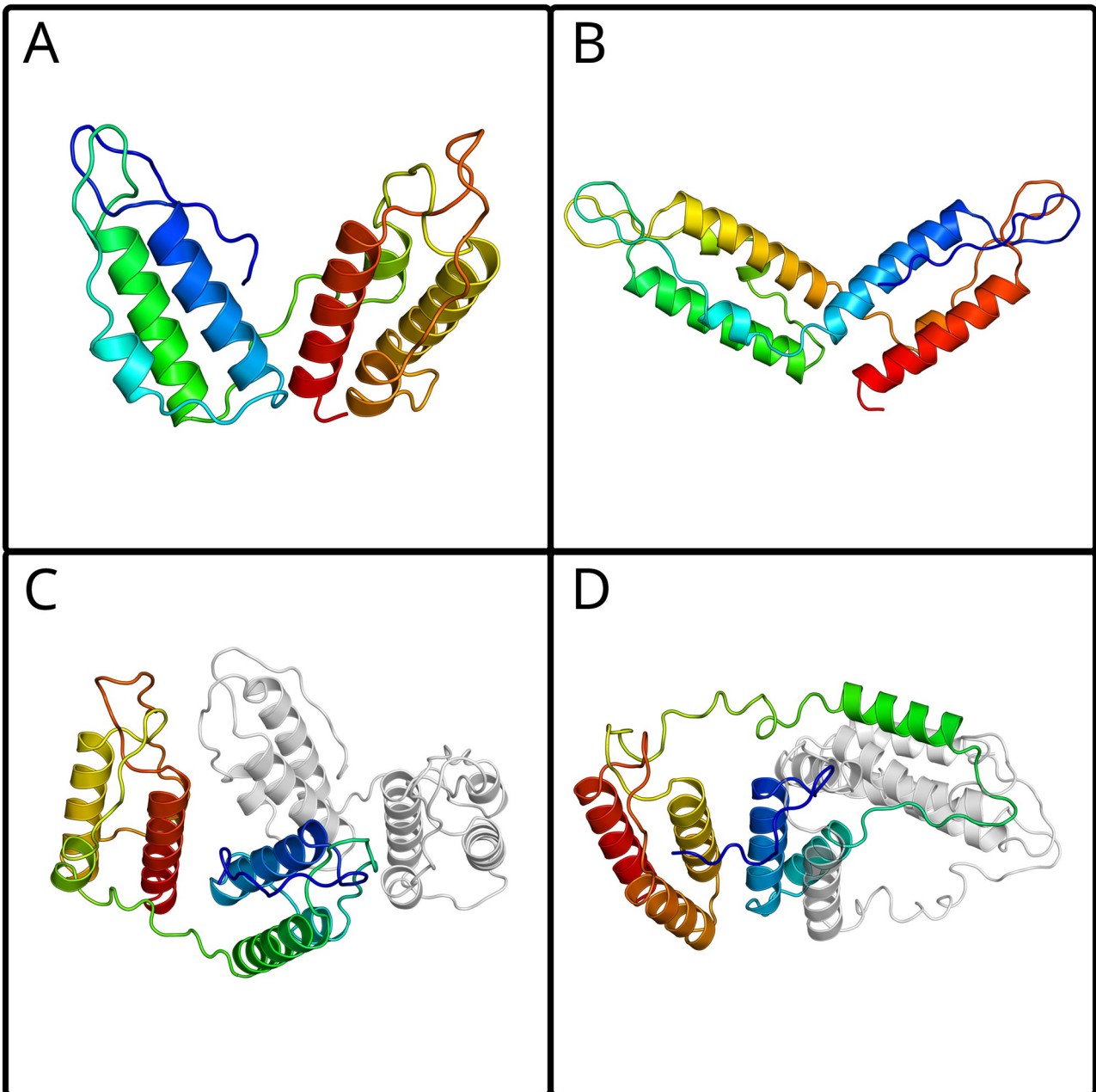

**Fig 3. Examples of the topologies observed in the PolyDIV AlphaFold models.** Colored from N-terminus (blue) to C-terminus (red). (A) The PURPLE_3_monomer model shows its DivIVA-like domains interacting in a consecutive manner, first with second and third with fourth. (B) The PURPLE_3+DivIVA_monomer shows a non-consecutive DivIVA-like topology. (C) PURPLE_3_dimer shows a topology equivalent to that of (A), where the two chains do not interact via their DivIVA-like domains. (D) PURPLE_1_dimer shows a complex topology where one or more DivIVA-like domains interact with another DivIVA-like domain from another polypeptide chain. All sequences and models are provided in the S1 File.

bacteria. GpsB is only found in Firmicutes, FilP only in Actinobacteria, and Scy exclusively in Streptomycetales (an order within Actinobacteria). It is well established in the literature that GpsB is a DivIVA paralog [2, 3], the product of an ancient duplication of the gene, and it has been noted that Scy and FilP are probably paralogs of each other as well [6]. Our discovery of a shared domain between all these proteins provides the first evidence for their homologous

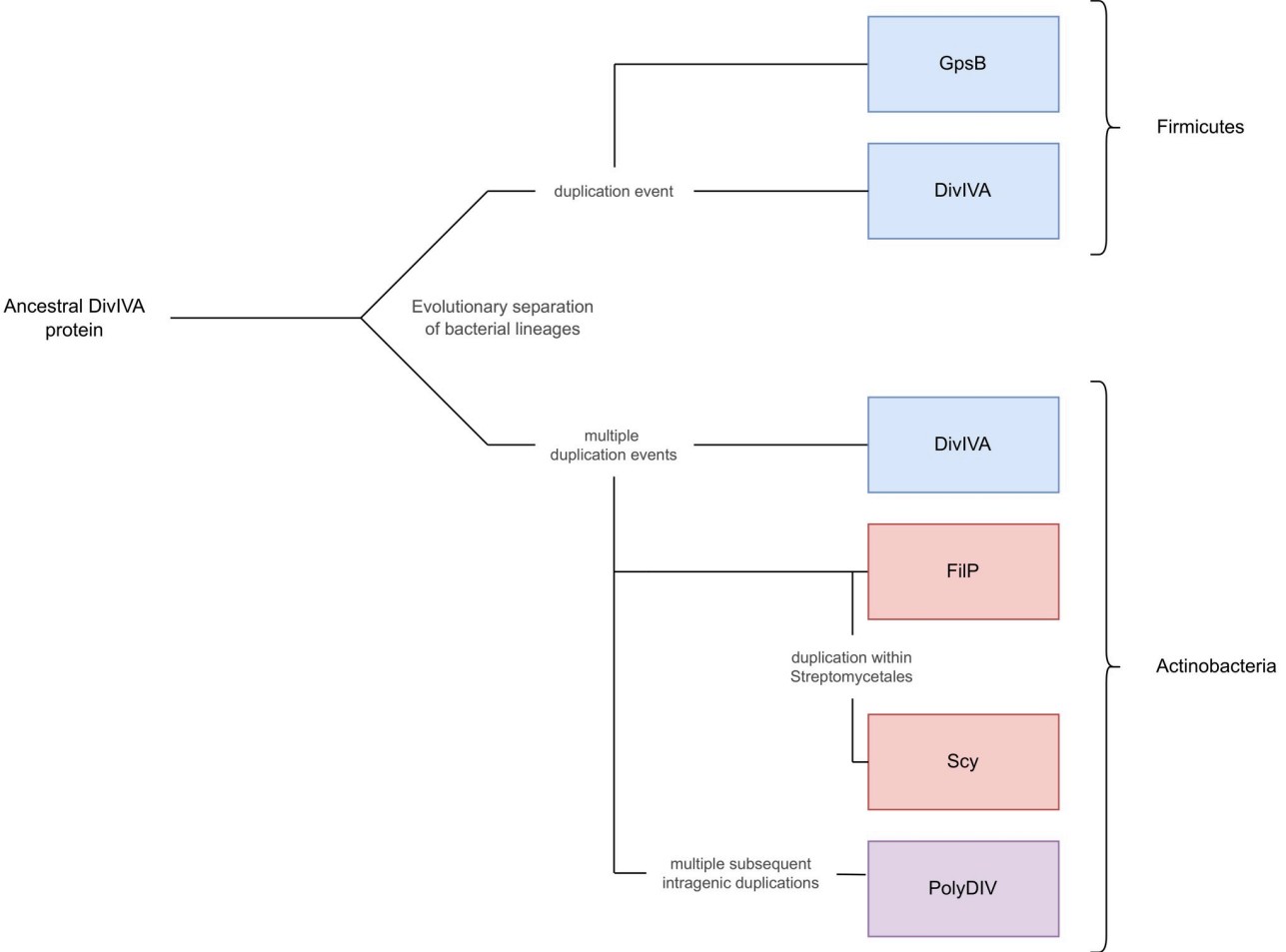

**Fig 4. Evolutionary scenario for proteins containing the DivIVA-like domain.** The diversity of the DivIVA-like superfamily seems to have arisen mainly through events of duplication (intra- or intergenic) and coiled-coil repeat expansion. The deepest branching point in this tree is an orthologous split due to the separation between Firmicutes and Actinobacteria. All other branch points are due to paralogy.

origin. We thus define here the new superfamily of DivIVA-like proteins and show that this includes a large group of unannotated proteins, the PolyDIVs. We propose that, like all other members of known function in this superfamily, PolyDIVs are also involved in cell division.

In evolutionary terms, it is clear that DivIVA is at the root of this superfamily, based on its broad taxonomic spectrum and its basic architecture. After the separation of Firmicutes and Actinobacteria within the Gram-positives, independent duplication events led to the origin of GpsB in the Firmicutes, and of FilP and PolyDIV in the Actinobacteria (Fig 4). While GpsB retained the basic architecture of DivIVA, FilP acquired a longer rod segment, in which the N-terminal heptad repeats were gradually expanded by hendecads, and PolyDIV underwent multiple intragenic duplications of the DivIVA domain, which changed the overall topology of the protein from a homodimer to a monomer. At a later stage, after the separation of Streptomycetes from the other Actinobacteria, a further duplication of FilP yielded Scy, which shows the longest coiled-coil rod among DivIVA-like proteins (at over 1000 residues) and is composed almost entirely of hendecads. This outlines the most likely evolutionary events that resulted in the modern complement of DivIVA-like proteins, but BLAST searches within individual proteomes provide many instances of DivIVA-like-related lineage-specific gene duplications (see

S2 Appendix in S1 File). These indicate that having multiple paralogs of one or the other of the DivIVA-like proteins is a widespread phenomenon and that the underlying duplications are an ongoing process.

Structurally, the FilP/Scy and the DivIVA/GpsB groups are obligate dimers. While our analyses suggest that PolyDIVs mostly fold as monomers, it is entirely possible that some oligomerize into more complex topologies. The advantage of the PolyDIV topology over the ancestral DivIVA one is unclear to us, but we note that PolyDIVs are widespread throughout Actinobacteria. Besides the evident increase in binding-site density along the same polypeptide chain and decreased requirements for folding and assembly, PolyDIVs may have the potential to act as hubs by displaying binding sites with different specificity in close proximity to each other. This, in turn, would explain why the DivIVA-like domains of PolyDIVs are more divergent compared to other DivIVA-like proteins.

The DivIVA, GpsB, FilP and Scy proteins are key players in the process of bacterial division. In spite of considerable effort, the mechanisms by which these proteins act within this process are incompletely understood and have not been synthesized into an overarching molecular model. Our identification of a domain conserved between these proteins and our observation that the highest sequence conservation is found in a loop displayed at the top of the domain immediately suggest that this is the main site for protein-protein interactions, providing a unifying feature for these otherwise diverse proteins.

## Supporting information

**S1 File. We provide S1 Fig (Series of CLANS maps at different P-value thresholds), S2 Appendix (Evidence of continuous genetic duplication events), and S3 Fig (Heatmap of pairwise sequence similarities for representative members of the DivIVA-like superfamily) as Supplementary information.** Furthermore, the following data are available at https://data. mendeley.com/datasets/bn627zbymx: (i) a cluster-map file, which can be navigated interactively in CLANS and gives direct access to all the sequences in this study, (ii) the sequence annotations for representatives of DivIVA-like proteins one for each cartoon shown in Fig 2B–2D, and (iii) the sequences and structural models of the PolyDIVs modeled with Alpha-Fold, both natural and artificial.
(DOCX)

## Acknowledgments

We would like to thank Dr Vikram Alva and Dr Stanislaw Dunin-Horkawicz for useful discussions and comments that contributed greatly to improve this manuscript.

The AlphaFold predictions were computed using the module available at the Raven cluster, part of the Max Planck Computing and Data Facility (https://www.mpcdf.mpg.de).

## Author Contributions

**Conceptualization:** Andrei N. Lupas.

**Funding acquisition:** Andrei N. Lupas.

**Investigation:** Mikel Martinez-Goikoetxea.

**Supervision:** Andrei N. Lupas.

**Writing – original draft:** Mikel Martinez-Goikoetxea.

**Writing – review & editing:** Mikel Martinez-Goikoetxea, Andrei N. Lupas.

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
