## [Decision Letter · Decision Letter 0]

15 Sep 2022

PONE-D-22-21666A conserved motif suggests a common origin for a group of proteins involved in the cell division of Gram-positive bacteriaPLOS ONE

Dear Dr. Mikel Martinez Goikoetxea,

Thank you for submitting your manuscript to PLOS ONE. After careful consideration, we feel that it has merit but does not fully meet PLOS ONE’s publication criteria as it currently stands. Therefore, we invite you to submit a revised version of the manuscript that addresses the points raised during the review process. In fact, as you can see the first Reviewer suggested rejection mainly because there are no experimental data to clearly support the conclusion. Despite I agree that such experimental data are needed to identify a new family of protein, I also believe that this bioinformatic and computational study '*is very likely to bring novel insights into our understanding of the evolution proteins involved in the cell division of Gram-positive bacteria*', as the second reviewer has pointed out. This second Reviewer suggested Minor Revisions.

Specifically, while more attention has to be given to the description of the procedures used for bioinformatic analyses (Reviewer 1, point 2-3), additional investigations are required to address specific questions and consolidate the conclusions (Reviewer 2). This will give more emphasis on the computational nature of the work, that can be highlighted in the title or abstract, and will increase the impact of the paper.

We look forward to receiving your revised manuscript.

Kind regards,

Matteo De March

Academic Editor

PLOS ONE

Journal Requirements:

Reviewers' comments:

Reviewer's Responses to Questions

**Comments to the Author**

1. Is the manuscript technically sound, and do the data support the conclusions?

Reviewer #1: No

Reviewer #2: Yes

2. Has the statistical analysis been performed appropriately and rigorously? 

Reviewer #1: N/A

Reviewer #2: Yes

3. Have the authors made all data underlying the findings in their manuscript fully available?

Reviewer #1: No

Reviewer #2: Yes

4. Is the manuscript presented in an intelligible fashion and written in standard English?

Reviewer #1: No

Reviewer #2: Yes

5. Review Comments to the Author

Reviewer #1: Goikoetxea M et.al. have conducted a study on proteins, involved in cell division DivIVA, GpsB, FilP, and Scy. A conserved N-terminal sequence motif has been identified, despite of their variability in occurrence & phenotypic profile. The motif, having coiled-coil helix structure is probably required for dimerization. They were collectively termed as DivIVA-like domain. The super secondary structure, consisting of two or more α-helices, wind around a central axis. The coiled coil structure, made by the repeats of seven residues, with a hydrophobic core. The hendecads repeats display similar properties. All this information has been procured by using various bioinformatics tool, like HMM-HMM, PFAM & HMMER. They MSA & BLAST of DivIVA, GpsB, FilP and Scy, and picked up homologues in other species.

However, the data obtained in this study is just obtained by computational tools, requires validation, and is not enough to accept for publication in present form. Thus, in my opinion it CAN NOT be accepted for publication.

Major Points-

1) All the information supported is not supported by experimental data. The results need experimental support.

2) There is no evolutionary tree or any figure to support the common origin of the proteins.

3) The study failed to explain how authors used GY[DN]xx[QE]V[DN] used to collect the sequence in NR3O database. How this pattern has been selected.

4) GpsB is mentioned as a paralog of DivIVA & Scy is mentioned as a paralog of FilP but no experimental proof has been mentioned to support the statement.

5) The results are not enough to identify a protein family. More experimental work is required.

Reviewer #2: The manuscript reports interesting observations regarding a a common origin for several families of proteins involved in the cell division of gram-positive bacteria. The identification of a new group of proteins (PolyDIV) whose role is not known experimentally. But by their affiliation to this family. The authors suggest that this new group of identified proteins is also involved in cell division. This last point is a hypothesis and its answer may be the subject of a future publication. This work is very likely to bring novel insights into our understanding of the evolution proteins involved in the cell division of Gram-positive bacteria but some parts of the study may require additional investigations to support the authors' message, to consolidate the main conclusions, and overall to increase the impact of the paper.

My main suggestions are:

I found this article interesting but I think the article, and especially the discussion would gain in impact and clarity if you would consider adding a phylogenetic figure to expose the different evolutionary steps described. A phylogenetic tree of the bacterias concerned, showing the nodes where each family of proteins described appear. Where are they present. Where are located the duplication events you are talking about. Are there also deletions in some clades, is it the result of horizontal transfer or evolutionary process by descent ect.

Minor comments:

Figure 1 panel C and D:

In the first column of the motif, there is a pink line that is probably the representation of the E and a white space below it. Why is the E not visible in small as in position 4 for example when there seems to be a white blank space in the representation. Moreover, in the text line 135 you say that the E is present in all DivIVA actinobacteria. its proportion is not just present in some rare sequences?

Position 8 appears to be a strict V and in your 1C sequence alignment there is an exception A at this position. Is this a very rare exception that results in a character of e.g., only 1 pixel on the 1D panel and therefore is not visible?

The use of AlphaFold, in spite of the impressive results of this software, should be described with more precaution by nuancing the results. For example, by adding a sentence in the results or in the discussion that the data remains a model that should be verified experimentally. Although it has a high chance of being correct for monomers. This precaution is especially important for more complex models like dimers and more.

6. PLOS authors have the option to publish the peer review history of their article (what does this mean?). If published, this will include your full peer review and any attached files.

Reviewer #1: No

Reviewer #2: No

---

## [Author Response · Author response to Decision Letter 0]

12 Dec 2022

Dear Editor,

thank you for the opportunity to submit a revised version of our article describing the new DivIVA-like domain. We have made changes to the manuscript in response to the comments we received from you and the reviewers, and we detail these changes in the following.

Reviewer 1

This reviewer provided the strong opinion in their general comments and in major points 1, 4 and 5 that experimental evidence would be needed to define a new protein domain. We disagree with this assessment both because of the standards in the sequence annotation field and our personal experience. Regarding the definition of new protein domains, we note that all protein domain databases in common use today (PFAM, SMART, CDD) were defined on the basis of sequence comparisons alone and the domains therein more than a decade before the first structures became available. In the structure classification databases (SCOP, CATH, ECOD) as well, homology, as inferred from sequence comparisons, is also the first classification criterium. These domains were mostly defined long before their functions could be ascertained, and for most domains recognized today, even the general biological activity continues to be elusive. We ourselves have contributed more than 50 such domains over the years, which are published and have become part of the domain databases. In this article we have done nothing else than apply the standards in the field to the definition of a new domain and we are unaware of any experimental approach which could help substantiate this definition.

In major point 2, the reviewer criticises the absence of a supporting evidence for the common origin of the DivIVA-like domains in different proteins. We are surprised by this comment because we have explained in detail in the paper the statistical criteria for inclusion of proteins in our cluster map (Figure 2) and we think that the BLAST p-value cutoff of 1E-15 that we used is extremely stringent. Nevertheless, we have now included in the supplementary material a heatmap of pairwise sequence similarities for representative members of the clusters we described, and hope that the reviewer will be convinced by the extremely strong signal of similarity.

In major point 3, the reviewer criticises the abscence of an explanation for how we obtained and used the pattern characteristic for DivIVA-like proteins. The generation of the pattern in the MEME program is explained in detail in section 2 of the Methods. The use of patterns in string searches is a basic aspect of informatics so we did not considered it necessary to detail this aspect of our work. We have therefore not further addressed this point in revision. 

Reviewer 2

In their main suggestion, the reviewer suggests that a phylogenetic figure summarizing the evolutionary steps proposed in our article would be a useful addition to the discussion. We agree and have included such a figure as the new Figure 4. At the level of resolution of our study, we are not aware of gene loss for any of these proteins that may have happened in specific lineages, nor of clear events of horizontal transfer superimposed on orthologous descent.

In the minor comments, the reviewer suggests potential problems with the representation of sequences in the alignment panel C versus motif panel D. We understand the points made, and agree that the way the figure legend was written was easy to misinterpret. We have therefore revised the figure legend, as well as the layout of panel C, in order to clarify the relevant points.

The reviewer also points out that some cautionary comments should be added to the paper concerning the use of AlphaFold models. We understand this point perfectly as well, all the more since we were assessors for the CASP14 experiment, and wish to point out two things in response: The first is that experimental structures are also models and have again and again produced erroneous outcomes due for example to crystal contacts, non-physiological buffer conditions, truncations needed for crystalization, and flawed model building (we note here the presence of several cis-prolines in CASP targets which were incorrectly modeled by the crystalographers and corrected after AlphaFold prediction). The second is that even for very complex CASP targets AlphaFold predictions were essential for a high-resolution model of datasets that could not otherwise be interpreted, showing that these predictions are complementary to experimental structure determination, not redundant with it. We have checked our manuscript carefully for instances that might be considered to overinterpret the meaning of the AlphaFold models, but we find that we have been quite cautious throughout.

We hope that with these changes and responses our manuscript is now fit for publication.

Sincerely,

Mikel Martinez Goikoetxea and Andrei N. Lupas

---

## [Decision Letter · Decision Letter 1]

2 Jan 2023

A conserved motif suggests a common origin for a group of proteins involved in the cell division of Gram-positive bacteria

PONE-D-22-21666R1

Dear M. Goikoetxea and A. Lupas,

We’re pleased to inform you that your manuscript has been judged scientifically suitable for publication and will be formally accepted for publication once it meets all outstanding technical requirements.

Kind regards,

Matteo De March

Academic Editor

PLOS ONE

Reviewers' comments:

Reviewer's Responses to Questions

**Comments to the Author**

1. If the authors have adequately addressed your comments raised in a previous round of review and you feel that this manuscript is now acceptable for publication, you may indicate that here to bypass the “Comments to the Author” section, enter your conflict of interest statement in the “Confidential to Editor” section, and submit your "Accept" recommendation.

Reviewer #2: All comments have been addressed

Reviewer #3: All comments have been addressed

2. Is the manuscript technically sound, and do the data support the conclusions?

Reviewer #2: Yes

Reviewer #3: Yes

3. Has the statistical analysis been performed appropriately and rigorously? 

Reviewer #2: Yes

Reviewer #3: Yes

4. Have the authors made all data underlying the findings in their manuscript fully available?

Reviewer #2: Yes

Reviewer #3: Yes

5. Is the manuscript presented in an intelligible fashion and written in standard English?

Reviewer #2: Yes

Reviewer #3: Yes

6. Review Comments to the Author

Reviewer #2: The various points raised have been argued by the author. And the requested changes have been taken into account correctly. The phylogenetic tree provides a real improvement in the overall understanding of the future reader.

Reviewer #3: Goikoetxea and Lupas presented an interesting conserved motif analysis focusing a common origin for a group of proteins involved in the cell division of Gram-positive bacteria. Considering the bottlenecks of a purely computational study, the authors have provided enough informatics based evidences to support their finding. I recommend this manuscript for publication.

7. PLOS authors have the option to publish the peer review history of their article (what does this mean?). If published, this will include your full peer review and any attached files.

Reviewer #2: No

Reviewer #3: **Yes: **Rahul Kaushik

---

## [Editor Report · Acceptance letter]

10 Jan 2023

PONE-D-22-21666R1 

A conserved motif suggests a common origin for a group of proteins involved in the cell division of Gram-positive bacteria 

Dear Dr. Martinez Goikoetxea:

I'm pleased to inform you that your manuscript has been deemed suitable for publication in PLOS ONE. Congratulations! Your manuscript is now with our production department. 

Kind regards, 

on behalf of

Dr. Matteo De March 

Academic Editor

PLOS ONE